# CascadeXML: Rethinking Transformers for End-to-end Multi-resolution Training in Extreme Multi-label Classification

**Siddhant Kharbanda**    **Atmadeep Banerjee**    **Erik Schultheis**    **Rohit Babbar**
Department of Computer Science, Aalto University, Finland
{firstname.lastname}@aalto.fi

## Abstract

Extreme Multi-label Text Classification (XMC) involves learning a classifier that can assign an input with a subset of most relevant labels from millions of label choices. Recent approaches, such as XR-Transformer and LightXML, leverage a transformer instance to achieve state-of-the-art performance. However, in this process, these approaches need to make various trade-offs between performance and computational requirements. A major shortcoming, as compared to the Bi-LSTM based AttentionXML, is that they fail to keep separate feature representations for each resolution in a label tree. We thus propose CascadeXML, an end-to-end multi-resolution learning pipeline, which can harness the multi-layered architecture of a transformer model for attending to different label resolutions with separate feature representations. CascadeXML significantly outperforms all existing approaches with non-trivial gains obtained on benchmark datasets consisting of up to three million labels. Code for CascadeXML will be made publicly available at `https://github.com/xmc-aalto/cascadexml`.

## 1  Introduction

Extreme multi-label text classification (XMC) deals with the problem of predicting the most relevant subset of labels from an enormously large label space spanning up to millions of labels. Over the years, extreme classification has found many applications in e-commerce, like product recommendation [7] and dynamic search advertisement [16], document tagging [1] and open-domain question answering [6, 26] and thus, practical solutions to this problem can have significant and far-reaching impact.

The output space in extreme classification is not only large but also, the distribution of instances among labels follows Zipf's law such that a large fraction of them are *tail labels* [2, 17]. For example, in a dataset of Wikipedia corpus containing 500K labels, only 2.1% labels annotate more than 100 data points and 60% of the labels annotate less than 10 data points! This inherently makes learning meaningful representations at an extreme scale a challenging problem. Further, as the memory and computational requirements grow linearly with the size of the label space, naive treatment via classical methods and off-the-shelf solvers fails to deal with such large output spaces [8, 54].

In extreme scenarios with millions of labels, a *surrogate* step of shortlisting candidate labels becomes crucial to perform extreme classification and has been adapted by many popular approaches [10, 19, 23, 52, 57]. For each data point, these approaches first solve a simpler task with coarse label-clusters or *meta-labels* as the label space to create a shortlist of relevant labels for the extreme classification task. This effectively reduces the training time and computational complexity of the extreme task to that of the shortlisting step which is $\mathcal{O}(\sqrt{L})$, where $L$ is the number of labels.

Existing works leverage label representation as a centroid of their annotated instances to create a Hierarchical Label Tree (HLT) [36, 55]. While earlier tree-based methods like Parabel [36] and

|  | AttentionXML | LightXML | XR-Transformer | CascadeXML |
|---|---|---|---|---|
| Base Model | Bi-LSTM | Transformer | Transformer | Transformer |
| Attention Maps | adaptive | shared | shared | adaptive |
| Single Encoder | × | ✓ | ✓ | ✓ |
| Multi-level HLT | ✓ | × | ✓ | ✓ |
| End-to-End | × | ✓ | × | ✓ |
| Full Resolution | ✓ | ✓ | × | ✓ |

Table 1: Strengths and limitations of current XMC methods: LightXML makes end-to-end training possible, but only admits a single level of the HLT. XR-Transformer allows for multiple levels, but its iterative feature learning does not train the transformer encoder at the highest resolution at all [57, Table 6]. However, by changing from multiple Bi-LSTMs to a single transformer, both LightXML and XR-Transformer lost AttentionXML's ability to adapt attention maps to each resolution.

Bonsai [22] used the entire HLT for extreme classification, many recent works [8, 10, 19, 23, 52] use label clusters only at a certain level of the HLT as meta-labels which are in turn used to shortlist candidate labels for the extreme task. In contrast, XR-Transformer [57] and AttentionXML [55] leverage multiple levels of the HLT, such that each level corresponds to a certain label *resolution* in the tree structure [57].

## 1.1 Motivation: Strengths and Weaknesses of Current Approaches

The earliest work to successfully combine label-tree based shortlisting and attention-based deep encoders is AttentionXML [55], which employs multiple Bi-LSTMs models that are trained sequentially for each tree resolution. More recent approaches [8, 19, 52, 57] replaced the model architecture with a more powerful transformer model [46] and fine-tune a pre-trained instance such as Bert [11]. Without careful designs, such models are computationally very expensive and earlier works [8, 52] could not effectively leverage transformers for both computation and performance on XMC tasks. Next, we discuss two contemporary works LightXML [19] and XR-Transformer [57].

**LightXML** introduces the concept of dynamic negative sampling, which replaces pre-computed label shortlists with a dynamically calculated shortlist that changes as the model's weights get updated. This enables end-to-end training with a single model by using the final feature representation of the transformer encoder for both the meta- and the extreme-classification task. Unfortunately, these two tasks appear to interfere with one another [23]. We hypothesize that this is because the meta task needs the attention maps to focus on different tokens than the extreme task. Furthermore, it only uses a single-level tree, which prevents scaling to the largest datasets.

**XR-Transformer** takes inspiration from multi-resolution approaches in computer vision like super resolution [25] and progressive growing of GANs [20, 21], and enables multiple resolutions through iterative training. However, unlike progressively grown GANs, which predict only at the highest resolution, XR-Transformer needs predictions across all resolutions for its progressive shortlisting pipeline, but uses representations trained at a single resolution. In practice, this leads to XR-Transformer having a complex multi-stage pipeline where the transformer model is iteratively trained up to a certain resolution and then frozen. This is followed by a *re-clustering* and *re-training* of multiple classifiers, working at different resolutions, with the same *fixed* transformer features [57, Alg. 2]. For datasets with over 500K labels, the transformer is only trained on up to $2^{15}$ clusters [57, Table 6], and the resulting features are used for the extreme task.

Unlike AttentionXML, using multiple instances of transformer models becomes undesirable due to their computational overhead. This enforces LightXML and XR-Transformer to make different trade-offs when leveraging a single transformer model for XMC tasks compared to AttentionXML, see Table 1. In this paper, we present a method, *CascadeXML*, that combines the strengths of these approaches, thus creating an end to-end trainable multi-resolution learning pipeline which trains a single transformer model across multiple resolutions in a way that allows the creation of label resolution specific attention maps and feature embeddings.

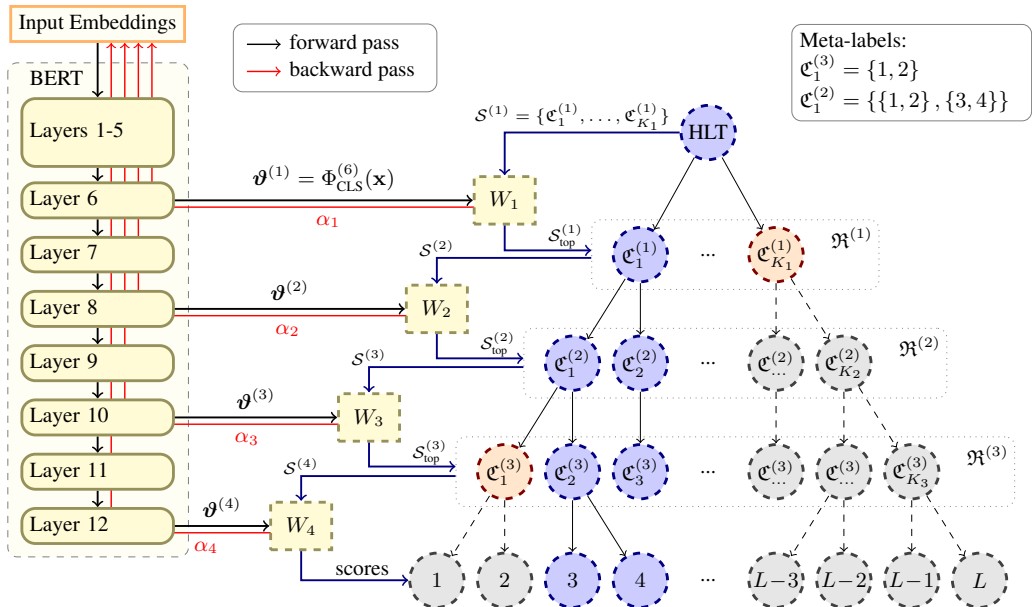

Figure 1: Overview over CascadeXML. The meta-classifiers $W_1$ - $W_3$ use intermediate BERT features $\boldsymbol{\vartheta}^{(t)}$ to discard all meta-labels except the highest scoring at that level (marked in red), so their descendants (gray) need not be considered at the next level. More details in § 3 and algorithm 1.

## 1.2 CascadeXML in a Nutshell

The key insights that enable calculating resolution-specific representations with a single forward pass through the transformer are **a)** that the shortlisting tasks get easier for lower label resolution and **b)** that the intermediate transformer representations are already quite powerful discriminators [49]. Consequently, CascadeXML extracts features for classification at the coarser meta-label resolution from earlier transformer layers and uses the last layer only for classification at the extreme resolution. The advantage of this is twofold: First, we postulate that the gain in classification accuracy due to enabling tree resolution-specific attention maps outweighs the loss in accuracy due to the slightly less powerful representations at the earlier transformer layers. Second, this ensures that the representation capacity of the later layers is exclusively available to the more difficult tasks.

The main idea of CascadeXML is illustrated in Figure 1. An input text is propagated through the transformer model until the first shortlisting level is reached. There, the embedding $\boldsymbol{\vartheta}^{(1)}$ corresponding to the [CLS] token is extracted, and used to determine scores for each level-1 meta-label. The highest-scoring meta-labels are selected to create a shortlist of candidate labels $\mathcal{S}_{\text{top}}^{(1)}$. The input is propagated further to perform another shortlisting step at the next resolution, at which a new [CLS] embedding $\boldsymbol{\vartheta}^{(2)}$ is extracted. Crucially, $\boldsymbol{\vartheta}^{(2)}$ is the result of a separate self-attention operation, and its attention map is a differently weighted combination of the token embeddings as compared to $\boldsymbol{\vartheta}^{(1)}$. The new embedding $\boldsymbol{\vartheta}^{(2)}$ is used to refine the shortlist by selecting the top-scoring level-2 meta-labels that are children of the shortlisted level-1 meta-labels. This process is repeated until at the final layer of the transformer, where classification is performed at the full label resolution.

**Contributions** In this paper, our contributions are the following:

- We introduce a novel paradigm which can harness the multi-layered architecture of transformers for learning representations at multiple resolutions in an HLT. This is in contrast to the existing approaches which treat transformer as merely a black box encoder,

- Implemented via CascadeXML, it is an end-to-end trainable multi-resolution learning pipeline which is quick to train and simple to implement. It eliminates the need of *bells and whistles* like multi-stage procedures, reclustering, and bootstrap training to achieve state-of-the-art performance, and

- CascadeXML not only scales to dataset with millions of labels but also improves the current state-of-the-art in terms of prediction performance by upto $\sim 6\%$ on benchmark datasets. In addition to this, it is computationally most efficient to train and clocks multi-fold improvement in inference time as compared to existing transformer-based approaches.

## 2 Notation & Preliminaries

**Problem setup:** In extreme classification, the input text instance (document or query) $\mathbf{x} \in \mathcal{X}$ is mapped to a subset of labels $\mathbf{y} \subset [L]$ out of the $L$ available labels identified through the integers $[L] := \{1, \ldots, L\}$, such that $L \sim 10^6$. Usually, we identify the labels through a binary vector $\mathbf{y} \in \{0, 1\}^L$, where $y_j = 1 \Leftrightarrow j \in \mathbf{y}$. The instances and labels are jointly distributed according to an unknown distribution $(\mathbf{x}, \mathbf{y}) \sim \mathbb{P}$ of which we have a i.i.d. sample $\mathcal{D} = \left\{ \left( \mathbf{x}^i, \mathbf{y}^i \right) : i \in [n] \right\}$ available as a training set. Even though the number of labels is large, the label vectors are sparse.

As part of the model pipeline, a text vectorizer $\Phi : \mathcal{X} \mapsto \mathbb{R}^d$ projects the documents to a $d$-dimensional embedding space. These vectorizers can be simple bag-of-words (BOW) or TF-IDF transformations $\Phi_{\text{tf-idf}}$, or a learnable model $\Phi_{\text{dnn}}$ that encodes the documents to the embedding space using weights learnt over the dataset. The final classification of an instance $\mathbf{x}$ is achieved by computing per label scores $\langle \mathbf{w}_l, \Phi(\mathbf{x}) \rangle$ using label-wise weight vectors $\mathbf{w}_l \in \mathbb{R}^d$ which are either (i) *jointly* learnt with $\Phi$ in a deep architecture [8, 10, 19, 32, 52, 55], or (ii) *solely* learnt in shallow classifiers [2, 56].

**Transformer Encoder:** More recent approaches in extreme classification have achieved state-of-the-art performance using pre-trained transformers as text vectorizers [19, 52, 57]. Popular transformer encoders for NLP [11, 28, 51] consist of an embedding layer, followed by a series of stacked self-attention layers and task-specific heads that can be fine-tuned along with the encoder. For a sequence of $m$ input words, each layer $a$ in these models produces a sequence of embeddings $\left( \Phi_0^{(a)}(\mathbf{x}), \ldots, \Phi_m^{(a)}(\mathbf{x}) \right)$, where $\Phi_i^{(a)}(\mathbf{x}) \in \mathbb{R}^d$ is a representation for the $i$'th token in the input sequence at the $a$'th layer, and $\Phi_{\text{CLS}}^{(a)}(\mathbf{x}) := \Phi_0^{(a)}(\mathbf{x})$ is an embedding corresponding to a special [CLS] token that has been pre-trained to capture a representation of the entire input [11, 28].

**Label Shortlisting:** Computing scores $\langle \mathbf{w}_l, \Phi(\mathbf{x}) \rangle$ for each label $l \in [L]$ in an extremely large output space is computationally very demanding, especially when the feature space $\mathbb{R}^d \ni \Phi(\mathbf{x})$ is high dimensional to allow for expressive features. To alleviate this problem, the labels $L$ are grouped together under $K$ *meta-labels* $\{\mathfrak{C}_1, \ldots, \mathfrak{C}_K\} =: \mathfrak{R}$, with $K \approx \mathcal{O}(\sqrt{L})$. Each meta-label is represented as the set of its child labels, $\mathfrak{C}_k = \{l \in [L] : l \text{ is child of } k\}$ and each target set $\mathbf{y} \subset [L]$ corresponds to a meta-target $\tilde{\mathbf{y}} = \mathfrak{R}(\mathbf{y}) := \{\mathfrak{C}_k \mid \exists i \in \mathbf{y} : i \in \mathfrak{C}_k\}$. A meta-label $\mathfrak{C}_l$ is relevant to a data point if it contains at least one relevant label.

Ideally, the labels grouped under one meta-label should be semantically similar to ensure the common representations learnt for meta- and extreme- tasks are relevant to both tasks [10]. This has the added advantage that the negative labels selected by the shortlisting procedure will be the most confusing ones, leading to hard negative sampling and improved learning of the extreme classifier [39].

The label vectors are sparse to the extent that the much coarser meta-labels still contain mostly zeros, $|\tilde{\mathbf{y}}| \ll K$. This means that it is enough to explicitly calculate the scores only for the descendants of relevant meta-labels. In practice, an algorithm will first predict scores for each meta-label and select the top-scoring ones as a *shortlist* for further classification. The meta- and extreme-classification tasks can either be learnt sequentially [8, 10] or jointly through dynamic label-shortlisting [19, 23].

For extremely large label spaces, a single label resolution for shortlisting is not optimal [57]. It is either too large that the meta-task itself becomes very expensive, or its individual clusters are so large that the shortlist contains too many candidate labels [57]. Thus, methods like AttentionXML and XR-Transformer use a multi-level shortlisting procedure based on a *Hierarchical Label Tree (HLT)*.

**Label Trees:** An HLT $\mathfrak{H} = \left\{ \mathfrak{R}^{(1)}, \ldots, \mathfrak{R}^{(T)} \right\}$ is a hierarchical clustering of labels into $T$ levels of successively refined clusters $\mathfrak{R}^{(t)} = \left\{ \mathfrak{C}_1^{(t)}, \ldots, \mathfrak{C}_{K_t}^{(t)} \right\}$. The clusters $\mathfrak{C}_k^{(t)}$ of each level form a disjoint partitioning of the clusters of the next, $\mathfrak{C}_k^{(t)} \subset \mathfrak{R}^{(t+1)}$, with the convention that the $T+1^{\text{th}}$ level is the full label space $\mathfrak{R}^{(T+1)} = [L]$. As such, each cluster $\mathfrak{C}_k^{(T)}$ (or meta-label) at level $T$ is represented by a set of labels, each cluster at level $T - 1$ is a set of level-$T$ meta-labels, and thus a set of sets

of labels, and so on. In practice, label representations $\mathbf{z}_l$ as an aggregate of their positive instance features are leveraged to create an HLT. Typically, $\mathbf{z}_l$ is constructed using $\Phi_{\text{tf-idf}}$ as

$$\mathbf{z}_l = \frac{\mathbf{v}_l}{\|\mathbf{v}_l\|}, \quad \text{where } \mathbf{v}_l = \sum_{i:\mathbf{y}_l^{(i)}=1} \Phi_{\text{tf-idf}}(\mathbf{x}^{(i)}).$$

The HLT is often constructed by recursively partitioning the label-space using balanced k-means clustering in a top-down fashion [8, 19, 55, 57]. For example, an HLT constructed using balanced 2-means clustering has $\lfloor \log_2 L \rfloor$ levels, out of which we select $T$ levels corresponding to different meta-label resolutions.

## 3   Method: CascadeXML

At its core, the CascadeXML model consists of three components: A pre-trained language model $\Phi$ which allows to extract a document representation $\Phi_{\text{CLS}}^{(a)}(\mathbf{x}) \in \mathbb{R}^d$ at different layers $a \in [A]$, an HLT $\mathfrak{H} = \{\mathfrak{R}^{(1)}, \ldots, \mathfrak{R}^{(T)}\}$ that divides the label space into $K_1 < \ldots < K_T < L = K_{T+1}$ successively refined clusters, and a set of linear classifiers $\left\{\mathbf{W}^{(t)} = [\mathbf{w}_1^{(t)}, \ldots, \mathbf{w}_{K_t}^{(t)}] \in \mathbb{R}^{d \times K_t} : t \in [T+1]\right\}$. For each meta-classifier, we select a layer in order to extract resolution-specific feature embeddings $\boldsymbol{\vartheta}^{(t)} := \Phi_{\text{CLS}}^{(a_t)}(\mathbf{x})$ with $1 < a_1 < \ldots < a_T < A = a_{T+1}$.

**Refining and Coarsening Label Vectors:** In order to transfer the representation of a shortlist of a label vector between different levels of the HLT, we need to define two operations. The first is refining the resolution of a shortlist: Given a shortlist $\mathcal{S} = \{\mathfrak{C}_{k_1}^{(t)}, \ldots, \mathfrak{C}_{k_s}^{(t)}\} \subset \mathfrak{R}^{(t)}$ of clusters of the resolution at level $t$, we want to calculate the representation of this shortlist in the next level, denoted $\mathfrak{R}_\downarrow^{(t)}(\mathcal{S})$. Since each cluster is defined as the set of its descendants, this is achieved by taking the union over the clusters in the list. The second operation is to find the cover of meta-labels that envelope a given set of labels, $\mathfrak{R}_\uparrow^{(t)}(\mathcal{S})$. This is achieved by identifying each of the clusters in the level above for which there exists at least one descendant in the shortlist. This leads to

$$\mathfrak{R}_\downarrow^{(t)}(\mathcal{S}) = \bigcup_{C \in \mathcal{S}} C \subset \mathfrak{R}^{(t+1)}, \qquad \mathfrak{R}_\uparrow^{(t)}(\mathcal{S}) = \left\{\mathfrak{C}_k^{(t-1)} \in \mathfrak{R}^{(t-1)} \mid \mathcal{S} \cap \mathfrak{C}_k^{(t-1)} \neq \emptyset\right\}. \tag{1}$$

We define $\mathfrak{R}^{(t)}(\mathbf{y}) = \left(\mathfrak{R}_\uparrow^{(t+1)} \circ \ldots \circ \mathfrak{R}_\uparrow^{(T)}\right)(\mathbf{y})$ to be the level-$t$ cover of the label set $\mathbf{y} \subset [L]$.

**Multi-resolution Dynamic Label Shortlisting:** We adapt dynamic label shortlisting [19] for multi-resolution training approaches. Formally, given a shortlist $\mathcal{S}^{(t)}$ of $s$ clusters $\{\mathfrak{C}_{k_1}^{(t)}, \ldots, \mathfrak{C}_{k_s}^{(t)}\} =: \mathcal{S}^{(t)} \subset \mathfrak{R}^{(t)}$ at level $t$, and the corresponding classification features $\boldsymbol{\vartheta}^{(t)}$, a new shortlist $\mathcal{S}^{(t+1)} \subset \mathfrak{R}^{(t+1)}$ is generated as follows: The $k$ highest-scoring meta-labels are selected into $\mathcal{S}_{\text{top}}^{(t)}$ and the next shortlist $\mathcal{S}^{(t+1)}$ is then given as the refinement:

$$\mathcal{S}_{\text{top}}^{(t)} = \left\{\mathfrak{C}_{k_j}^{(t)} : j \in \text{Top}_k\left(\left\langle \mathbf{w}_{k_i}^{(t)}, \boldsymbol{\vartheta}^{(t)} \right\rangle\right)\right\}, \quad \mathcal{S}^{(t+1)} = \mathfrak{R}_\downarrow^{(t)}\left(\mathcal{S}_{\text{top}}^{(t)}\right) \tag{2}$$

At the first level, however, the shortlist is initialized to consider all meta labels $\mathcal{S}^{(1)} = \mathfrak{R}^{(1)}$. It is of essence that all true positive (meta-)labels appear at every level during training time [8, 19, 55]. However, this is difficult to ensure implicitly due to imperfect recall rate of classifiers. Hence, during training all meta-labels corresponding to ground-truth labels are added to the candidate set, leading to

$$\mathcal{S}_{\text{train}}^{(t+1)} = \mathfrak{R}_\downarrow^{(t)}(\mathcal{S}_{\text{top}}^{(t)}) \cup \mathfrak{R}^{(t+1)}(\mathbf{y}). \tag{3}$$

Consequently, a single training step in CascadeXML looks as given in algorithm 1.

**Training Objective:** At each level of the tree, the goal is to correctly identify the most likely meta-labels, which can be achieved through minimization of a one-vs-all loss, such as BCE

$$\mathcal{L}^{(t)}(\mathbf{x}, \mathbf{y}) = \frac{1}{|\mathcal{S}^{(t)}|} \sum_{l \in \mathcal{S}^{(t)}} \mathcal{L}^{(t)}\left(\langle \mathbf{w}_{k_l}^{(t)}, \boldsymbol{\vartheta}^{(t)}(\mathbf{x})\rangle, \mathfrak{R}^{(t)}(\mathbf{y})_l\right). \tag{4}$$

**Algorithm 1:** Training step in CASCADEXML

---

**Input:** instance $\mathbf{x}$, labels $\mathbf{y}$, clusters $\mathfrak{C}$, features $\Phi$, classifiers $\mathbf{W}$, weights $\boldsymbol{\alpha}$

$\boldsymbol{\vartheta}^{(t)} \leftarrow \Phi_{\text{CLS}}^{(t)}(\mathbf{x}) \; \forall t \in [T+1]$      /* Forward pass through Transformer */

$\text{loss} \leftarrow 0$

$\mathcal{S}^{(1)} \leftarrow \mathfrak{C}^{(1)}$      /* Initialize shortlist */

**for** $t$ *in* $1, 2, ..., T$ **do**

     $\mathbf{y}^{(t)} \leftarrow \mathfrak{R}^{(t)}(\mathbf{y})$      /* Ground-truth meta-labels */

     $p_i \leftarrow \langle \mathbf{w}_{k_i}^{(t)}, \boldsymbol{\vartheta}^{(t)} \rangle \; \forall i \in \mathcal{S}^{(t)}$      /* Sparse prediction */

     $\text{loss} \leftarrow \text{loss} + \frac{\alpha_t}{|\mathcal{S}^{(t)}|} \sum_{l \in \mathcal{S}^{(t)}} \mathcal{L}^{(t)}\left(p_l, y_l^{(t)}\right)$

     $\mathcal{S}_{\text{top}}^{(t)} \leftarrow \left\{ \mathfrak{C}_{k_j}^{(t)} : j \in \text{Top}_k\left(\{p_i \mid i \in \mathcal{S}^{(t)}\}\right) \right\}.$      /* Top-K selection */

     $\mathcal{S}^{(t+1)} \leftarrow \mathfrak{R}_{\downarrow}^{(t)}(\mathcal{S}_{\text{top}}^{(t)}) \cup \mathfrak{R}^{(t+1)}(\mathbf{y})$      /* Add ground-truth positives */

$p_i \leftarrow \langle \mathbf{w}_{k_i}^{(T+1)}, \boldsymbol{\vartheta}^{(T+1)} \rangle \quad \forall i \in \mathcal{S}^{(T+1)}$      /* Final prediction */

$\text{loss} \leftarrow \text{loss} + \frac{\alpha_{T+1}}{|\mathcal{S}^{(T+1)}|} \sum_{l \in \mathcal{S}} \mathcal{L}^{(T+1)}(p_l, y_l)$

adjust $\Phi$ and $\mathbf{W}$ to reduce loss

---

The overall objective is a weighted sum of the individual layer losses

$$\mathcal{L}(\mathbf{x}, \mathbf{y}) = \sum_{t=1}^{T+1} \alpha^{(t)} \mathcal{L}^{(t)}(\mathbf{x}, \mathbf{y}), \tag{5}$$

where level $T+1$ corresponds to the extreme classification task. To balance out different loss scales as a result of normalization by varied shortlist sizes, we re-scale losses from multiple resolutions using a simple factor $\alpha^{(t)} = |\mathcal{S}^{(t)}| / \min_{t \in [T+1]}(|\mathcal{S}^{(t)}|)$.

**Inference:** The algorithm for inference remains the same as training, except we do not teacher-force true (meta-)labels $\mathfrak{R}^{(t)}(\mathbf{y})$ during inference time (Eqn. 3). The inference consists of one forward pass through the transformer backbone, followed by successive shortlisting at all $T$ resolutions, and final extreme classification. For a $b$-way branching tree with $\log_b L$ levels, each level $t$ has $b^t$ meta-labels, of which $|\mathcal{S}^{(t)}| \propto \sqrt{b^t}$ are selected for the shortlist. Therefore, the time complexity is

$$\mathcal{O}(\mathcal{T}_{\text{dnn}} + (|\mathcal{S}^{(1)}| + |\mathcal{S}^{(2)}| + ... + |\mathcal{S}^{(T+1)}|)|d) = \mathcal{O}(\mathcal{T}_{\text{dnn}} + (\sum_{t=1}^{\log_b L} \sqrt{b^t})d) = \mathcal{O}(\mathcal{T}_{\text{dnn}} + d\sqrt{L})$$

where $\mathcal{T}_{\text{dnn}}$ denotes the time taken to compute transformer embeddings for an input.

## 4 Main results

**Benchmark datasets & Evaluation Metrics** We benchmark the performance of CascadeXML on standard datasets and evaluation metrics, as shown in Table 2 and Table 3. In all, we use 5 public XMC datasets [4, 31], the statistics of which are specified in appendix A.1. For all our experiments we use the same raw text input, train-test split and sparse feature representations as used in [55, 19, 57]. We use Precision@k (P@k) and Propensity-scored P@k (PSP@k) [17], which focuses more on the model's performance on tail labels, as the evaluation metrics.

**Baseline Methods** We compare CascadeXML with strong baselines: DiSMEC [2], Parabel [36], eXtremeText [47], Bonsai [22], XR-Linear [56], AttentionXML, X-Transformer [8], LightXML and XR-Transformer. The baseline results for these methods were obtained from [57, Table: 2]. It may be noted that, in order to incorporate the global statistical information of the dataset, XR-Transformer concatenates sparse *tf-idf* features to those learnt via deep networks (denoted $\Phi_{\text{dnn}}$) as:

$$\Phi_{\text{cat}}(\mathbf{x}) = \left[ \frac{\Phi_{\text{dnn}}(\mathbf{x})}{\|\Phi_{\text{dnn}}(\mathbf{x})\|}, \frac{\Phi_{\text{tf-idf}}(\mathbf{x})}{\|\Phi_{\text{tf-idf}}(\mathbf{x})\|} \right]$$

For fair evaluation, we present separate comparison with methods which use (i) features from a DNN encoder only, and (ii) those obtained by combining DNN and tf-idf features.

**Model Ensemble Settings** For CascadeXML, we follow the ensemble settings of LightXML and XR-Transformer. Specifically, we ensemble a model each using BERT [11], RoBerta. [28] and XLNet [51] for Wiki10-31K and AmazonCat-13K. For the larger datasets, we make an ensemble of three BERT models with different seeds for random initialization. On the other hand, AttentionXML uses ensemble of 3 models and X-Transformer uses 9 model ensemble with BERT, RoBerta, XLNet large models with three difference indexers.

| Metrics | P@1 | P@3 | P@5 | P@1 | P@3 | P@5 | P@1 | P@3 | P@5 |
|---|---|---|---|---|---|---|---|---|---|
| Datasets | **Wiki-500K** | | | **Amazon-670K** | | | **Amazon-3M** | | |
| DiSMEC | 70.21 | 50.57 | 39.68 | 44.78 | 39.72 | 36.17 | 47.34 | 44.96 | 42.80 |
| Parabel* | 68.70 | 49.57 | 38.64 | 44.91 | 39.77 | 35.98 | 47.42 | 44.66 | 42.55 |
| eXtremeText | 65.17 | 46.32 | 36.15 | 42.54 | 37.93 | 34.63 | 42.20 | 39.28 | 37.24 |
| Bonsai* | 69.26 | 49.80 | 38.83 | 45.58 | 40.39 | 36.60 | 48.45 | 45.65 | 43.49 |
| XR-Linear | 65.59 | 46.72 | 36.46 | 43.38 | 38.40 | 34.77 | 47.40 | 44.15 | 41.87 |
| | DNN Features | | | | | | | | |
| AttentionXML* | 76.74 | 58.18 | 45.95 | 47.68 | 42.70 | 38.99 | 50.86 | 48.00 | 45.82 |
| LightXML* | 77.89 | 58.98 | 45.71 | 49.32 | 44.17 | 40.25 | - | - | - |
| CascadeXML* | **78.39** | **59.86** | **46.49** | **50.22** | **45.20** | **41.45** | **52.75** | **50.83** | **48.90** |
| | [DNN $\oplus$ tf-idf] Features | | | | | | | | |
| X-Transformer** | 77.09 | 57.51 | 45.28 | 48.07 | 42.96 | 39.12 | 51.20 | 47.81 | 45.07 |
| XR-Transformer* | 79.40 | 59.02 | 46.25 | 50.11 | 44.56 | 40.64 | **54.20** | 50.81 | 48.26 |
| CascadeXML* | **81.13** | **62.43** | **49.12** | **52.15** | **46.54** | **42.44** | 53.91 | **51.24** | **49.52** |

Table 2: Comparison of CascadeXML to state-of-the-art methods on large scale benchmark datasets. '*' and '**' imply ensemble results of 3 models and 9 models respectively, and '-' implies that the model does not scale for that dataset due to GPU memory constraints.

**Empirical Performance:** As demonstrated in Table 2, CascadeXML, which leverages the multi-layered transformer architecture to learn label resolution specific feature representations, is empirically superior to all existing XMC approaches whether using: (i) tf-idf features, or (ii) DNN-based dense features, or (iii) both. This is despite CascadeXML being a simple end-to-end trainable algorithm without using any bells or whistles like multi-stage procedures [55, 57], reclustering [57] or bootstrapped training [55, 57]. For example, on Amazon-3M dataset, CascadeXML significantly outperforms all DNN-based approaches while taking only 24 hours to train on a single Nvidia A100 GPU. In contrast, X-Transformer needs to ensemble 9 models and 23 days to train on 8 GPUs to reach the empirical performance (c.f. page 2, [57]), and LightXML remains unscalable.

We report results on Wiki10-31K and AmazonCat-13K in Table 3. As a standard practice in the domain [6, 19, 57], we use 256 tokens as an input sequence length to the model after truncation. We find this sequence length to be sufficient for competent empirical performance and hence, we do not witness any empirical benefits of leverage sparse tf-idf features for these datasets. Notably, CascadeXML achieves best results on AmazonCat-13K dataset and performs at par with XR-Transformer without leveraging sparse tf-idf features. Thus, CascadeXML's unique multi-resolution approach is a favorable choice not only for datasets with extremely large output spaces, but also for datasets with tens of thousand labels.

| Metrics | P@1 | P@3 | P@5 | P@1 | P@3 | P@5 |
|---|---|---|---|---|---|---|
| Datasets | **Wiki10-31K** | | | **AmazonCat-13K** | | |
| DiSMEC | 84.13 | 74.72 | 65.94 | 93.81 | 79.08 | 64.06 |
| Parabel | 84.19 | 72.46 | 63.37 | 93.02 | 79.14 | 64.51 |
| eXtremeText | 83.66 | 73.28 | 64.51 | 92.50 | 78.12 | 63.51 |
| Bonsai | 84.52 | 73.76 | 64.69 | 92.98 | 79.13 | 64.46 |
| XR-Linear | 85.75 | 75.79 | 66.69 | 94.64 | 79.98 | 64.79 |
| | DNN Features | | | | | |
| AttentionXML* | 87.34 | 78.18 | 69.07 | 95.84 | 82.39 | 67.32 |
| LightXML* | 89.67 | 79.06 | 69.87 | 96.77 | 83.98 | 68.63 |
| CascadeXML* | **89.74** | 80.13 | 70.75 | **96.90** | **84.13** | **68.78** |
| | [DNN $\oplus$ tf-idf] Features | | | | | |
| X-Transformer** | 88.26 | 78.51 | 69.68 | 96.48 | 83.41 | 68.19 |
| XR-Transformer* | 88.69 | **80.17** | 70.91 | 96.79 | 83.66 | 68.04 |
| CascadeXML* | 89.18 | 79.71 | **71.19** | 96.71 | 84.07 | 68.69 |

Table 3: Comparison of CascadeXML to state-of-the-art methods on legacy datasets.

On the concatenated feature space, CascadeXML also significantly outperforms XR-Transformer on 8 out of 9 dataset-metric combination (ref: Table 2) with notable $\sim 6\%$ improvement on Wiki-500K dataset for P@5 metric. As XR-Tranformer learns transformer features for label resolutions only up to $2^{15}$ clusters [57, Table 6], incorporating tf-idf features via a computation-intensive XR-Linear [56] for final extreme classification task becomes an integral part of their pipeline. On the other hand, CascadeXML does not have any such inherent limitation, and even without tf-idf features, it outperforms XR-Transformer on 7 out of 9 dataset-metric combinations (ref: Table 2).

## 5   Computational cost & Ablation results

**Single Model Comparison & Training Time**   Empirical results and training time of a single CascadeXML model has been compared to single instance performance of state-of-the-art DNN based XMC approaches in Table 4. Notably, CascadeXML, without leveraging any sparse tf-idf features, performs at par with XR-Transformer which benefits from using the extra statistical information about the entire (not truncated) data point. When comparing training time, CascadeXML is up to an order of magnitude faster than LightXML on a single GPU. In multi-GPU setting, CascadeXML trains in 7.2 hours on Wiki-500K dataset using only 4 NVidia V100 GPUs as compared to XR-Transformer and AttentionXML which require 12.5 hours on 8 GPUs. CascadeXML not only reduces the training time across datasets but also requires half the number of GPUs to do so. This proves that CascadeXML is significantly more compute efficient as compared to previous DNN based XMC approaches.

| Dataset | Method | P@1 | P@3 | P@5 | $T_{\text{train}}^1$ | $T_{\text{train}}^m$ |
|---|---|---|---|---|---|---|
| Wiki10-31K | AttentionXML | 87.1 | 77.8 | 68.8 | - | 0.5 |
| | LightXML | 87.8 | 77.3 | 68.0 | 6.7 | - |
| | XR-Transformer (+ tf-idf) | 88.0 | 78.7 | 69.1 | 1.3 | 0.5 |
| | CascadeXML | 88.4 | 78.3 | 68.9 | **0.3** | **0.1** |
| Wiki-500K | AttentionXML | 75.1 | 56.5 | 44.4 | - | 12.5 |
| | LightXML | 76.3 | 57.3 | 44.2 | 89.6 | - |
| | XR-Transformer (+ tf-idf) | 78.1 | 57.6 | 45.0 | 29.2 | 12.5 |
| | CascadeXML | 77.0 | 58.3 | 45.1 | **22.0** | **7.2** |
| Amazon-670K | AttentionXML | 45.7 | 40.7 | 36.9 | - | 8.1 |
| | LightXML | 47.3 | 42.2 | 38.5 | 53.0 | - |
| | XR-Transformer (+ tf-idf) | 49.1 | 43.8 | 40.0 | 8.1 | 3.4 |
| | CascadeXML | 48.5 | 43.7 | 40.0 | **7.5** | **3.0** |

Table 4: Single model comparison of DNN based XMC approaches. $T_{train}^1$ denotes the training time as reported on a single GPU (Nvidia V100). $T_{train}^m$ denotes multi-GPU training time on 8 GPUs for AttentionXML and XR-Transformer, and on only 4 GPUs for CascadeXML.

**Inference Time**   CascadeXML clocks the fastest inference speed as compared to all previous DNN-based XMC models, as shown in Table 5. Notably, CascadeXML is $\sim 1.5\times$ and $\sim 2\times$ faster at inference time compared to LightXML and XR-Transformer respectively.

| Dataset | AttentionXML | X-Transformer | XR-Transformer | LightXML | CascadeXML |
|---|---|---|---|---|---|
| Wiki10-31K | 20.0 | 48.1 | 39.1 | 27.1 | **12.8** |
| AmazonCat-13K | 14.4 | 47.6 | 26.1 | 24.1 | **13.1** |
| Wiki-500K | 80.1* | 48.1 | 33.9 | 27.3 | **16.0** |
| Amazon-670K | 76.0* | 48.0 | 30.9 | 23.3 | **16.6** |
| Amazon-3M | 130.5* | 50.2 | 35.2 | - | **16.9** |

Table 5: Comparison of CascadeXML w.r.t. inference time (in milliseconds per sample) with SOTA XMC methods. Inference times were recorded on a single Nvidia V100 GPU and a single CPU with a batch size of 1. Superscript * implies that model parallel was used with 8 GPUs for inference.

**Performance on Tail Labels**   Performance of CascadeXML on tail labels has been compared to baseline XMC methods in Table 6. We note that for smaller datasets - Wiki10-31K and AmazonCat-13K - PfastreXML [17] significantly outperforms all other methods on PSP metrics. This is to be expected, as this method specifically optimizes for tail labels. However, for larger datasets - Amazon-670K and Wiki-500K - CascadeXML outperforms the strong baselines like XR-Transformer

and AttentionXML by 5-8% on PSP@3 and PSP@5 metrics while also significantly outperforming PfastreXML over these datasets. These results conclude that our unique end-to-end multi-resolution training approach is not only empirically superior to previous approaches in P@K metrics but also outperforms strong XMC baselines in performance over tail labels.

| Methods | PSP@1 | PSP@3 | PSP@5 | PSP@1 | PSP@3 | PSP@5 | PSP@1 | PSP@3 | PSP@5 | PSP@1 | PSP@3 | PSP@5 |
|---------|-------|-------|-------|-------|-------|-------|-------|-------|-------|-------|-------|-------|
| | **Wiki10-31K** | | | **AmazonCat-13K** | | | **Amazon-670K** | | | **Wiki-500K** | | |
| DiSMEC | 10.60 | 12.37 | 13.61 | 51.41 | 61.02 | 65.86 | 26.26 | 30.14 | 33.89 | 27.42 | 32.95 | 36.95 |
| ProXML | 17.17 | 16.07 | 16.38 | 61.92 | 66.93 | 68.36 | 30.31 | 32.31 | 34.43 | - | - | - |
| PfastreXML | **19.02** | **18.34** | **18.43** | 69.52 | **73.22** | 75.48 | 29.30 | 30.80 | 32.43 | 32.02 | 29.75 | 30.19 |
| Parabel | 11.69 | 12.47 | 13.14 | 50.92 | 64.00 | 72.10 | 26.36 | 29.95 | 33.17 | 26.88 | 31.96 | 35.26 |
| Bonsai | 11.85 | 13.44 | 14.75 | 51.30 | 64.60 | 72.48 | 27.08 | 30.79 | 34.11 | 27.46 | 32.25 | 35.48 |
| | DNN Features (Single Model) | | | | | | | | | | | |
| XML-CNN | 9.39 | 10.00 | 10.20 | 52.42 | 62.83 | 67.10 | 17.43 | 21.66 | 24.42 | - | - | - |
| AttentionXML | 16.20 | 17.05 | 17.93 | 53.52 | 68.73 | 76.26 | 29.30 | 32.36 | 35.12 | 30.05 | 37.31 | 41.74 |
| XR-Transformer | 12.16 | 14.86 | 16.40 | 50.51 | 64.92 | 74.63 | 29.21 | 33.49 | 37.65 | 32.10 | 39.41 | 43.75 |
| CascadeXML | 13.22 | 14.70 | 16.10 | 52.08 | 67.46 | 76.19 | 30.23 | 34.93 | 38.79 | 31.25 | 39.35 | 43.29 |
| | DNN Features (Ensemble Model) | | | | | | | | | | | |
| AttentionXML* | 15.57 | 16.80 | 17.82 | 53.76 | 68.72 | 76.38 | 30.29 | 33.85 | 37.13 | 30.85 | 39.23 | 44.34 |
| CascadeXML* | 13.36 | 15.06 | 16.56 | 52.68 | 68.50 | 77.52 | **31.19** | **36.07** | 40.15 | 32.60 | 42.03 | 46.66 |
| | [DNN $\bigoplus$ tf-idf] Features (Ensemble Model) | | | | | | | | | | | |
| XR-Transformer* | 12.25 | 15.00 | 16.75 | 50.72 | 65.66 | 75.95 | 29.77 | 34.05 | 38.29 | **32.70** | 40.44 | 45.02 |
| CascadeXML* | 13.32 | 15.35 | 17.45 | 51.39 | 66.81 | **77.58** | 30.77 | 35.78 | **40.52** | 32.12 | **43.15** | **49.37** |

Table 6: Comparison of performance of state-of-the-art methods on tail labels on benchmark datasets.

**Impact of label clusters size** Earlier works have argued that using fine-grained clusters leads to better model performance [19, 23, 32]. With increased number of label clusters at the penultimate level of HLT, the multi-resolution tasks tend to get more in-sync with each other [23] and hence enables transfer learning across different stages of learning pipelines [10, 32]. Our mutli-resolution training enables CascadeXML to use $2^{16}$ label clusters for penultimate label resolution, double that of XR-Transformer. As Table 7 shows, increasing the resolution at the last shortlisting step results in non-trivial improvements, highlighting the need for architectures that can efficiently handle high-resolution meta-labels.

| Dataset | Model Setting | P@1 | P@3 | P@5 |
|---------|---------------|-----|-----|-----|
| Amazon-670K | Default | **48.5** | **43.7** | **40.0** |
| | $2^{15}$ Clusters | 47.6 | 42.9 | 39.3 |
| | w/o Rescaling | 47.8 | 42.8 | 39.2 |
| Wiki-500K | Default | **76.9** | **58.4** | **45.2** |
| | $2^{15}$ Clusters | 76.6 | 58.1 | 44.9 |
| | w/o Rescaling | 76.5 | 57.9 | 44.7 |
| Amazon-3M | Default | **51.3** | **49.0** | **46.9** |
| | $2^{15}$ Clusters | 50.8 | 48.7 | 46.5 |
| | w/o Rescaling | 50.9 | 48.6 | 46.6 |

Table 7: Ablation experiments showing performance of a *single* CascadeXML model with different model settings. Default implies $2^{16}$ clusters in penultimate label resolution and per-resolution loss re-scaling.

**Impact of re-scaling per-resolution loss** For extreme datasets, we witness more efficient training by having a larger shortlist size for finer label resolutions (see appendix A.2.2). This results in varied loss scales across label resolutions leading to one resolution dominating the training. Comparing the "Default" and "w/o Rescaling" rows for respective datasets in Table 7, indicates that rescaling as introduced in section 3 helps CascadeXML train more effectively.

## 6 Discussion

As shown above, CascadeXML substantially outperforms current state-of-the-art approaches both in terms of computational efficiency, and prediction performance. It achieves this while being simpler than XR-Transformer and more scalable than LightXML. Revisiting Table 1, the extraction of intermediate [CLS] representations enables us to perform end-to-end multi-resolution training with resolution-specific attention maps from a single transformer model. From Table 2, we show improvements over both LightXML and XR-Transformer, which are transformer-based models sharing feature representations across multiple label resolutions. We attribute this increased performance to CascadeXML's end-to-end multi-resolution training pipeline and property of adapted feature embed-

dings. The presented architecture, therefore, combines the strength of different previous approaches, without inheriting their limitations.

We postulated that the (meta-)classification tasks at different resolutions need to attend to different tokens in the input sequence, a property lacking in recent approaches [8, 19, 57]. This intuition is corroborated by the fact that the attention maps learned by CascadeXML differ significantly between different resolutions. We provide graphs and supporting data in Appendix B.

Obtaining label resolution specific feature embeddings in a pre-trained transformer model requires to make the compromise of extracting these at earlier transformer layers, where the token representations are not yet as refined as in the final layer [49]. In contrast to the extreme classification task, which requires placing true labels exactly at the top-5 positions, the shortlisting task only requires all true meta-labels be recalled within the shortlist, which is much longer than five elements. We observe that the meta-classifiers are able to achieve high-recall (as shown Appendix B) at the shortlist length even with the less refined features. CascadeXML is, therefore, successful in efficiently leveraging intermediate transformer representations for the task of label shortlisting while keeping feature representations separate for each label resolution.

The structure of the model means that earlier layers in the transformer receive gradients from multiple resolutions, whereas the final layers' back propagation is limited to gradients from the extreme resolution (cf. Figure 1). This setup ensures that a portion of the representational capabilities is exclusively available for the extreme task and hence, the final transformer layers are increasingly more suitable for classification at the extreme resolution. At the same time, since the gradients from the finer tasks are also relevant for the coarser meta-label shortlisting task, so additional downstream task does not hurt the shortlisting performance at the intermediate transformer layers.

## 7   Other Related Works

A large majority of the initial works in XMC have been focussed on learning the classifier (with fixed features), with one of the following class of methods : (i) one-vs-rest [54, 2, 53, 3, 41], (ii) label trees [35, 22, 47, 18], (iii) decision trees [37, 44, 30], and (iv) label embedding based [5, 45, 14] classifiers. Beyond the above algorithmic categorization of approaches, computational considerations on scaling the training process via negative sampling [16], and smart initialization [12, 41] has also been studied. Furthermore, the statistical consequences of negative sampling [39] and missing labels [17, 38, 43, 42, 48] have led to novel insights in designing unbiased loss functions.

With advances in deep learning, joint learning of features and classifier has been the focus of most recent approaches. Building on the first steps for developing convolutional neural networks for text classification [24], deep extreme classification was introduced in XML-CNN [27]. This work preceded the recent developments on employing BiLSTM and transformer encoders discussed in detail the Introduction section. There has been a recent surge in works to tackle XMC for instances with short text inputs and those in which labels are endowed with textual descriptions [10, 9, 32, 33, 40]. Lately, the XMC setting has also been extended to predict unseen labels in zero-shot learning [15, 58, 50].

## 8   Conclusion

In this paper, we introduced a novel paradigm for fine-tuning transformer models in XMC settings. In contrast to the existing methods, which consider transformer as a blackbox encoder, we leverage its multi-layered architecture to learn data representation corresponding to different resolutions in the HLT as well as fine-grained labels at the level of the extreme classifier. The proposed instantiation of our framework in the form of CascadeXML not only yields state-of-the-art prediction performance, but is also end-to-end trainable (without intermediate reclustering steps), simpler to implement, and fast on training and inference. Beyond further research in extreme classification towards fully harnessing the representation capabilities of transformer encoders, we believe our approach can inspire future multi-resolution architectures in other domains as well which leverage label hierarchy.

## 9   Acknowledgments

The authors would like to thank Devaansh Gupta and Mohammadreza Qaraei for useful discussions. They also acknowledge the support of CSC – IT Center for Science, Finland, as well as the Aalto Science-IT project, for providing the required computational resources. This research is supported in part by Academy of Finland grants : Decision No. 348215 and 347707.

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
