# A Appendix

## A.1 Dataset Details & Evaluation Metrics

As stated earlier, the main application of Extreme Multi-label Text Classification is in e-commerce - product recommendation and dynamic search advertisement - and in document tagging, where the objective of an algorithm is to correctly recommend/advertise among the top-k slots. Thus, for evaluation of the methods, we use precision at $k$ (denoted by $P@k$), and its propensity scored variant (denoted by $PSP@k$) [17]. These are standard and widely used metrics by the XMC community [4]. For each test sample with observed ground truth label vector $y \in \{0,1\}^L$ and predicted vector $\hat{y} \in \mathbb{R}^L$, $P@k$ is given by :

$$\mathrm{P}@k(\mathbf{y}, \hat{\mathbf{y}}) := \frac{1}{k} \sum_{\ell \,\in\, \mathrm{top}@k(\hat{y})} y_\ell$$

where $\mathrm{top}@k(\hat{y})$ returns the $k$ largest indices of $\hat{y}$.

Since $P@k$ treats all the labels equally, it doesn't reveal the performance of the model on tail labels. However, because of the long-tailed distribution in XMC datasets, one of the main challenges is to predict tail labels correctly, which may be more valuable and informative compared to head classes. A set of metrics that have been established in XMC to evaluate tail performance are propensity-scored version of precision. These $PSP@k$ were introduced in [17], and use a weighting factor based on a propensity score $p_\ell$ to give more weight to tail labels:

$$\mathrm{PSP}@k(\mathbf{y}, \hat{\mathbf{y}}) := \frac{1}{k} \sum_{\ell \,\in\, \mathrm{top}@k(\hat{y})} \frac{y_\ell}{p_\ell}.$$

We use the empirical values for $p_\ell$ as proposed in [17].

| Datasets | $\mathbf{d}_{\text{tf-idf}}$ | # Labels | # Training | # Test | ALpP | APpL |
|---|---|---|---|---|---|---|
| **Wiki10-31K** | 101,938 | 30,938 | 14,146 | 6,616 | 18.64 | 8.52 |
| **AmazonCat-13K** | 203,882 | 13,330 | 1,186,239 | 306,782 | 5.04 | 448.57 |
| **Wiki-500K** | 2,381,304 | 501,070 | 1,779,881 | 769,421 | 4.75 | 16.86 |
| **Amazon-670K** | 135,909 | 670,091 | 490,449 | 153,025 | 5.45 | 3.99 |
| **Amazon-3M** | 337,067 | 2,812,281 | 1,717,899 | 742,507 | 36.04 | 22.02 |

Table 8: Dataset Statistics. APpL denotes the average data points per label, ALpP the average number of labels per point. For a fair comparison with other baselines, we download these five publicly available benchmark datasets from https://github.com/yourh/AttentionXML.

## A.2 Model Details

| Datasets | Transformer Layer : Label Resolutions | Shortlist Size | Ep | $N_x$ | Dropouts |
|---|---|---|---|---|---|
| **Wiki10-31K** | $\{9, 10\} : 2^9 \mid 11 : 2^{12} \mid 12 : 30938$ | $2^9, 2^9, \sim 2^9$ | 15 | 256 | 0.3, 0.3, 0.4 |
| **AmazonCat-13K** | $\{7, 8\} : 2^8 \mid 10 : 2^{11} \mid 12:13330$ | $2^8, 2^8, \sim 2^8$ | 6 | 256 | 0.2, 0.3, 0.4 |
| **Wiki-500K** | $\{5, 6\} : 2^{10} \mid 8 : 2^{13} \mid 10 : 2^{16} \mid 12 : 501070$ | $2^{10}, 2^{10}, 2^{11}, \sim 2^{12}$ | 12 | 128 | 0.2, 0.25, 0.35, 0.5 |
| **Amazon-670K** | $\{5, 6\} : 2^{10} \mid 8 : 2^{13} \mid 10 : 2^{16} \mid 12 : 670091$ | $2^{10}, 2^{10}, 2^{11}, \sim 2^{12}$ | 15 | 128 | 0.2, 0.25, 0.4, 0.5 |
| **Amazon-3M** | $\{5, 6\} : 2^{10} \mid 8 : 2^{13} \mid 10 : 2^{16} \mid 12 : 2812281$ | $2^{10}, 2^{10}, 2^{11}, \sim 2^{12}$ | 15 | 128 | 0.2, 0.25, 0.3, 0.5 |

Table 9: Hyperparameters for CascadeXML. Ep denotes the total number of epochs needed to fine-tune model over the dataset. $N_x$ is the number of text tokens input to the model after truncation. Transformer layers put in brackets next to the label resolution imply that a concatenation of the [CLS] token of the respective layers has been used for label shortlisting at that resolution.

### A.2.1 Model Hyperparameters

CascadeXML optimizes the training objective using Binary Cross Entropy loss as the loss function and AdamW [29] as the optimizer. We use different learning rates for the transformer encoder and the (meta-)label weight vectors as we need to train the weight vectors from random initialization in

contrast to fine-tuning the transformer encoder. Specifically, the transformer encoder is fine-tuned at a learning rate of $10^{-4}$, while the weight vectors are trained at a learning rate of $10^{-3}$. The learning rate schedule consists of a constant learning rate for most of the iterations, with a cosine warm-up at the beginning and cosine annealing towards the end of the schedule. In multi-GPU setting, we use a batch size of 64 per GPU (256 total) across 4 GPUs. In single GPU setting, we still use a batch size of 256 by accumulating gradients for 4 iterations. The hyperparameter settings in detail have been mentioned in Table 9.

### A.2.2 Ensemble Training Time

As shown in Table 10, CascadeXML achieves the lowest training time across all datasets except Amazon-3M using only 4 GPUs as compared to XR-Transformer which leverages 8 GPUs. Note that on Amazon-3M, XR-Transformer achieves a slightly lower training time. However, XR-Transformer uses $2\times$ the number of GPUs and does not train the DNN model on full 3 million label resolution. XR-Transformer trains the DNN model using a classification training object comprising of only $2^{15}$ label clusters [57, Table 7] and then leverages XR-Linear [56], a linear solver, to scale up to 3M labels. On the other hand, CascadeXML trains an ensemble of 3 models to full resolution of 3 million labels in 30 hours using only 4 GPUs.

| Dataset | AttentionXML-3 | X-Transformer-9 | LightXML-3 | XR-Transformer-3 | CascadeXML-3 |
|---|---|---|---|---|---|
| Wiki10-31K | 1.5 | 14.1 | 26.9 | 1.5 | **0.4** |
| AmazonCat-13K | 24.3 | 147.6 | 310.6 | 13.2 | **9.8** |
| Wiki-500K | 37.6 | 557.1 | 271.3 | 38.0 | **21.6** |
| Amazon-670K | 24.2 | 514.8 | 159.0 | 10.5 | **9.0** |
| Amazon-3M | 54.8 | 542.0 | - | **29.3** | 30.0 |

Table 10: Time taken to train the ensembles of the respective models. Training time AttentionXML, X-Transformer and XR-Transformer have been reported using 8 NVidia V100 GPUs. Training time for LightXML is clocked using 1 GPU and that of CascadeXML is clocked using 4 GPUs.

### A.3 Leveraging Sparse Features

As we are using BERT for the transformer backbone of our method, we have to truncate the input sequences to a limited number of tokens (see Table 9). This truncation results in loss of information. Thus, following XR-Transformer's lead we combine the features trained by CasadeXML with statistical information in the form of sparse tf-idf representation of the full input in an additional OVA classifier. The concatenated features are constructed as [19]:

$$\Phi_{\text{cat}}(\mathbf{x}) = \left[ \frac{\Phi_{\text{dnn}}(\mathbf{x})}{\|\Phi_{\text{dnn}}(\mathbf{x})\|}, \frac{\Phi_{\text{tf-idf}}(\mathbf{x})}{\|\Phi_{\text{tf-idf}}(\mathbf{x})\|} \right]$$

We use a version of DiSMEC [2] instead of using XR-Linear [56] - as done in XR-Transformer - as our external OVA classifier for $\Phi_{\text{cat}}$. Even though XR-Linear achieves slightly better performance than DiSMEC across datasets (Table: 1), we find DiSMEC to be more resource efficient than XR-Linear. To quantify, DiSMEC runs successfully on 116GB RAM for all datasets, while XR-Linear requires close to 470GB RAM for Amazon-3M. Next, we discuss the application of DiSMEC over $\Phi_{\text{cat}}$.

**DiSMEC** DiSMEC is a linear multilabel classifier that minimizes an $L_2$-regularized squared hinge-loss, followed by a pruning step to only keep the most important weights. Thus, the loss for a given weight matrix $\mathbf{W} = [\mathbf{w}_1, \ldots, \mathbf{w}_L]$ is given by

$$\mathcal{L}[\mathbf{W}] = \lambda \|\mathbf{W}\|_2^2 + \sum_{i=1}^{N} \sum_{j=1}^{L} \max\left(0, 1 - y_{ij}\langle\Phi_{\text{cat}}(\mathbf{x}_i), \mathbf{w}_j\rangle\right)^2. \tag{6}$$

Crucially, from the point of view of the OVA classifiers, the input features $\Phi_{\text{cat}}(\mathbf{x}_i)$ are constant. This means that the task decomposes into independent optimization problems for each label, minimizing

$$\mathcal{L}[\mathbf{w}_j] = \lambda \|\mathbf{w}_j\|_2^2 + \sum_{i=1}^{N} \max\left(0, 1 - y_{ij}\langle\Phi_{\text{cat}}(\mathbf{x}_i), \mathbf{w}_j\rangle\right)^2. \tag{7}$$

This allows for trivial parallelization of the task across CPU cores, and also means that the weights $\mathbf{w}_j$ can be pruned as soon as the sub-problem is solved. Consequently, there is no need to ever store the entire weight matrix, improving memory efficiency.

The objective function (7) has a continuous derivative, and its Hessian is well defined everywhere except exactly at the decision boundary. Consequently, it can be minimized using a second-order Newton method. A discussion of this in the context of linear classification can be found in [13].

## B   Visualizations and Analysis

In this section we provide visualizations and additional data that corroborate our interpretation that different attention- and feature maps are needed for classification at different granularities of the label tree. In Figure 2, the attention of the [CLS] token to itself in the previous layer is visualized.

If this self-attention were large, then $\Phi_{\mathrm{CLS}}^{(t)}$ would be mostly a function of $\Phi_{\mathrm{CLS}}^{(t-1)}$. In such a case, $\Phi_{\mathrm{CLS}}^{(t-1)}$ would contain less information than $\Phi_{\mathrm{CLS}}^{(t)}$ (data-processing inequality), but the meta-labels $\mathfrak{R}^{(t-1)}(\mathbf{y})$ at level $t-1$ contain strictly less information than $\mathfrak{R}^{(t)}(\mathbf{y})$. Thus, is the [CLS] token embedding had strong feed-forward characteristics, $\Phi_{\mathrm{CLS}}^{(t-1)}$ would have to contain all the information also about the extreme-level labels, and thus have limited representation capacity for the level-t task.

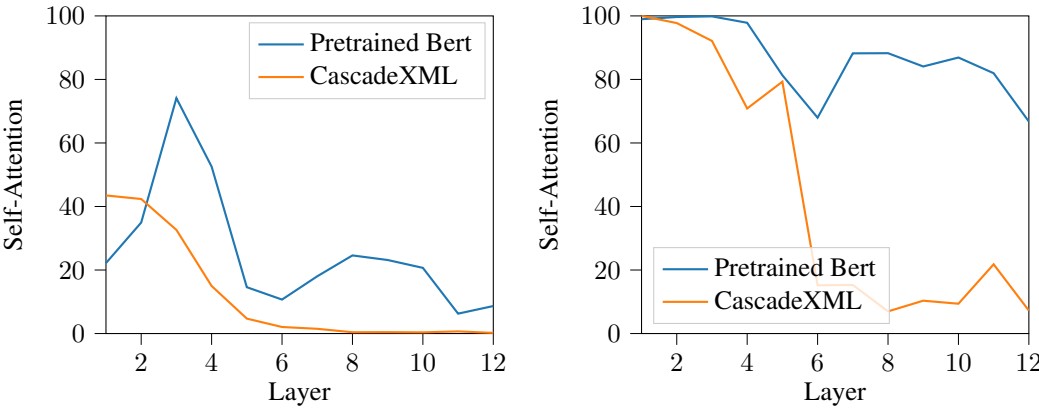

Figure 2: Average (left) and maximum (right) self-attention of the [CLS] token at a given layer to the [CLS] token in the preceding layer. The embedding for the [CLS] token is almost exclusively assembled from the embeddings of the other tokens in the later layers much more strongly in CascadeXML than in pretrained BERT.

Luckily, Figure 2 is a sanity-check that shows that this is not the case. Starting from layer 6, where the first meta-task is placed, the [CLS] token is almost entirely re-assembled at each layer from the embeddings of the other tokens – much more strongly than in a pretrained BERT. This allows each layer to extract the information best suited for classification at the given hierarchy level.

We can also detect some qualitative differences in the attention maps at different resolutions: The entropy, i.e. how much the attention is concentrated or spread across different tokens, changes significantly between levels. This is not an artifact of the pretrained BERT model, but appears to be learned during fine-tuning.

In Figure 4, we analyse the flow of information and visualize how much processing is happening in a given layer with respect to the [CLS] token. We rely on projection weighted canonical correlation analysis [34] for this task. This allows to compare the representations at different layers in a way that is invariant to any linear transformations.

We primarily make two observations from Figure 4. First, we observe that [CLS] token embeddings of layers 6, 8, 10 and 12 are more closely related in CascadeXML than in LightXML. This is expected as LigthXML only trains the (meta-)label weight vectors using the [CLS] token embeddings of the bottom layers. Because the multi-resolution training objectives differ only in granularity, many

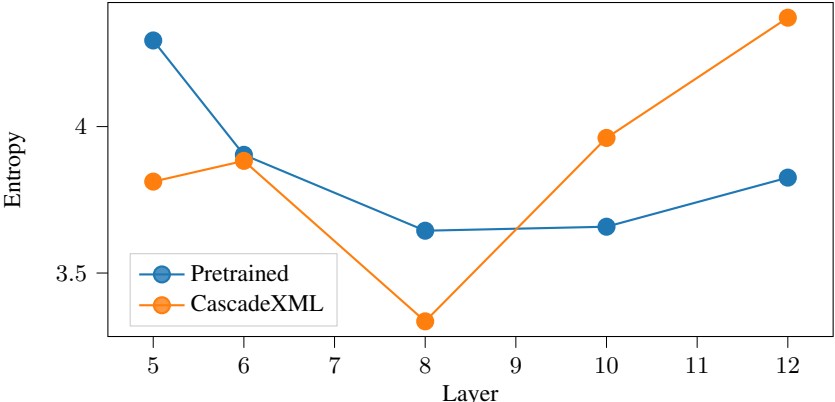

Figure 3: Entropy of the distribution of attention to the input tokens. A large value indicates that attention is given to many different tokens, whereas a smaller value means that few tokens receive most of the attention.

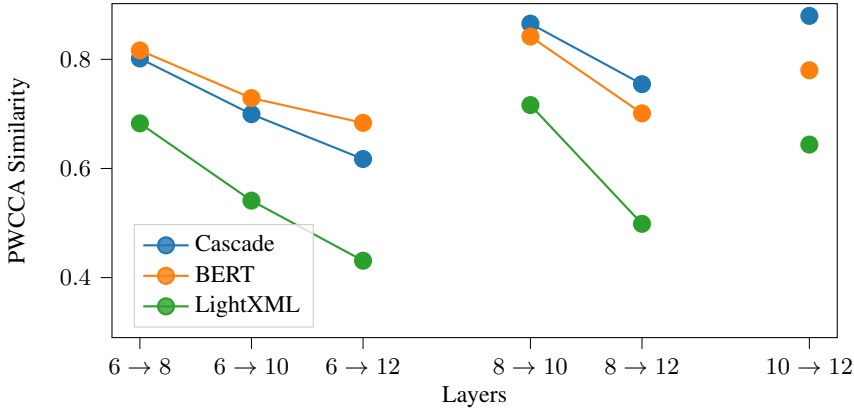

Figure 4: PWCCA similarity between [CLS] token representations at different levels for CascadeXML trained on Amazon-670K dataset. For this dataset, we place the weight vectors of different label resolutions at layers 6, 8, 10 and 12.

features required to distinguish coarse meta-labels are useful in determining finer meta-labels as well. Hence, the similarity between consecutive representations is expected to be strong. On the other hand, when only looking at CascadeXML's points (in blue) in Figure 4, we observe that the tasks in the first meta-classifier and the extreme classifier are substantially different. This implies both training objectives require different representations that cannot be provided by a LightXML-/XR-Transformer-style model which use same attention maps (and hence, same [CLS] token embeddings) for all resolutions.

Table 11: Recall of the shortlisting tasks.

| Dataset | Level 1 | Level 2 | Level 3 |
| --- | --- | --- | --- |
| Amazon-670K | 98.29 | 91.93 | 83.1 |
| Wiki-500L | 99.54 | 96.53 | 93.12 |

The ability to use earlier layers' [CLS] token representation for the meta-task crucially depends on the fact that these representations are still sufficient for achieving high recall in the shortlisting task. As shown in Table 11, the shortlisting achieves very good recall rates. In particular, the very first shortlist, with the "weakest" features, achieves almost perfect recall.