# OpenReview forum: "CascadeXML: Rethinking Transformers for End-to-end Multi-resolution Training in Extreme Multi-label Classification"
_NeurIPS.cc/2022/Conference — NeurIPS 2022 Accept_

### Official Review · Reviewer_zXmp · 2022-07-06

**Rating:** 5
**Confidence:** 5
**Soundness:** 2 fair
**Presentation:** 3 good
**Contribution:** 2 fair

**Summary:**

This work proposes CascadeXML which can leverage the multi-layered architecture to learn data representation corresponding to different resolutions in the Hierarchical Label Tree as well as fine-grained labels at the level of the extreme classifier. Each layer of the model can perform label shortlisting on a level.

The authors compare the proposed CascadeXML to strong baselines on three XMC benchmark datasets. Experiments show that the proposed method outperforms these baselines, and can achieve faster training and inference.


**Questions:**

1. Can you explain more on Eq(1)? How can we present a shortlist with its descendants, which might be a set of another shortlists? How can we get the corresponding vector representations?

2. If any prediction error happens on a higher level during inference, will this error propagate easily to lower levels?

**Limitations:**

The authors have discussed and addressed some of the concerns I have in the paper.

**Strengths And Weaknesses:**

Strength:

1. The proposed CascadeXML is end-to-end differentiable and achieves better results compared to previous baselines on three benchmark datasets.

2. The proposed method is faster in both training and inference, which is an important aspect of the XMC task.

3. This work is well-written and easy to follow.


Weakness:

1. The idea of dynamic label shortlisting is very interesting, but I wonder how to solve the cases where HLT contains more levels than the layer number of Transformers? Also, for the HLT containing fewer levels, do we have to change the model structure? If the model is trained on HLT-1 containing $n$ layers, can we transfer it to an HLT with $m$ layers where $m \neq n$?

2. Why do the earlier layers in the Transformer correspond to the higher levels of the HLT? More explanations over this may help a bit with the answer to Q1.

3. The proposed method highly relies on the multi-layer structure of the Transformer, which makes it not easy to extend to other frameworks. As mentioned above, the scalability and transferability of this method may be much more limited compared to the baselines adopted in this work.

---

> ### Author Response · Authors · 2022-08-02
> **response to reviewer zXmp**
>
> 1. **HLT levels and Transformer layers** : Please check the common response on the correspondence between transformer layers and the levels in HLT. Transferring the weights from the model trained one HLT to another is possible. This can be done by simply changing the resolution objective and reinitializing the meta-classifier at that transformer layer. In practice, this should help in better initialization than an off-the-shelf pretrained implementation of BERT (check Appendix B).
>
> 2. **Earlier layers in Transformer and top levels in HLT** : We explain this intuition in section 1.2.
>
> 3. **Extension to other frameworks** : We do not really understand what the reviewer means by “other frameworks”. Our method works for a range of transformer models and has been tried for BERT, XLNet and RoBERTa. Further, it is at least as scalable as most Bert based models for XMC. Given the more intermediary layers in our HLT than those in LightXML (which has only one meta-classification layer and also uses a Bert encoder), CascadeXML scales to Amazon3M dataset with 3 Million labels, while LightXML cannot.
>
> 4. **More details on Equation (1)** : Consider the example visualized in Figure 1.
> At the first level, the shortlist is $\mathcal{S}^{(1)} = \lbrace\mathfrak{C}\_1^{(1)}\rbrace$. $\mathfrak{C}\_1^{(1)}$ in itself is defined as the union of its descendants, so $\mathfrak{C}\_1^{(1)} = \lbrace \mathfrak{C}\_1^{(2)}, \mathfrak{C}\_2^{(2)} \rbrace$.
> Thus, by applying (1) we get $\mathcal{S}^{(2)} = \mathfrak{R}\_{\downarrow}^{(1)}(\mathcal{S}^{(1)}) = \lbrace\mathfrak{C}\_1^{(2)}, \mathfrak{C}\_2^{(2)}\rbrace$. Looking at the figure again, we see $\mathfrak{C}_1^{(2)} = \lbrace\mathfrak{C}_1^{(3)}, \mathfrak{C}_2^{(3)}\rbrace$ and $\mathfrak{C}\_2^{(2)} = \lbrace\mathfrak{C}\_3^{(3)}\rbrace$. Therefore,  $\mathfrak{R}\_{\downarrow}^{(2)}(\mathcal{S}^{(2)}) = \lbrace\mathfrak{C}\_1^{(3)}, \mathfrak{C}\_2^{(3)}, \mathfrak{C}\_3^{(3)}\rbrace$.
>
> For the other direction, assume that 3 and $L-1$ were the only relevant labels, i.e. $\mathsf{y} = \lbrace3, L-1\rbrace$.
> Then, by applying (1) we
> get $\mathfrak{R}\_{\uparrow}^{(1)} = \lbrace\mathfrak{C}\_2^{(3)}, \mathfrak{C}\_{K\_3}^{(3)}\rbrace$, because $3 \in \mathfrak{C}\_1^{(3)} = {3, 4}$
> and $L-1 \in \mathfrak{C}\_{K\_3}^{(3)} = \lbrace L-1, L\rbrace$.
>
> 5. **Controlling classification error at top-levels** : Instead of making a single top-most prediction at higher levels, the prediction procedure (as in training) involves shortlisting candidate meta-labels (also known as beam-search). This leads to much higher accuracy (and hence high recall of the actual labels, see Table 7 in Appendix B) at the top levels of the HLT. To ensure a high recall, we select the shortlist size appropriately.

---

> > ### Author Response · Authors · 2022-08-08
> > **Further clarification**
> >
> > Dear reviewer zXmp,
> >
> > Hope you had a chance to look at our rebuttal above. Furthermore, we would like to add an additional point that was missed out in earlier rebuttal.
> >
> > 6. **Number of benchmark datasets** : The proposed approach CascadeXML has been evaluated on **five** (not just 3, as suggested by the reviewer in the first point under strengths heading) benchmark datasets. The main paper presents three datasets and results on the two more datasets (Wiki10-31K and AmazonCat-13K) are given in the Appendix A.3.1. Also, this has been mentioned in Lines 197-200 of the main paper.
> >
> > We would appreciate your feedback in response to rebuttal and hope it will further improve your assessment score of the paper.  We are glad to answer any further questions.

---

> > > ### Comment · Reviewer_zXmp · 2022-08-08
> > > **Rebuttal response**
> > >
> > > Thanks for the responses. I think the responses do not fully address my concern in the HLT layer number and the Transformer model layer number. It is still not clear to me what will happen if there are more HLT layers than the Transformer model layer. I would like to see more explanations or experiments on how to initialize the model in a further revision. Therefore, I would like to keep the original score.

---

> > > > ### Author Response · Authors · 2022-08-08
> > > > **Number of HLT levels**
> > > >
> > > > Dear reviewer,
> > > >
> > > > many thanks for your comments.
> > > >
> > > > Regarding your above question, please find our response below :
> > > >
> > > > **Number of HLT levels** : While constructing the HLT, the number of levels is a hyper-parameter, and it can be set to a reasonably value as desired. Larger values i.e. O(log L) resulting from binary partition were last used in Parabel (Prabhu et, al. 2018). Owing to error propagation with large number of HLT levels in Parabel, most works, since then, have restricted to using 3 or 4 levels. This includes --
> > > > 1. AttentionXML (You et al. NeurIPS 2019)
> > > > 2. Bonsai (Khandagale et al. MLJ 2020)
> > > > 3. X-Transformer (Chang et al. KDD 2020)
> > > > 4. LightXML (Jiang et al. AAAI 2021)
> > > > 5. XR-Transformer (Zhang et al. NeurIPS 2021)
> > > >
> > > > **Shallow HLT construction** : Shallow HLT is constructed by first creating a binary HLT and then selecting levels from the same. This is the approach taken by CascadeXML, along with the above mentioned works. This process can also be thought of as creating a m-ary tree at each of those chosen levels. For examples, throughout our experiments, our created HLT can be viewed as an 8-ary tree from each level to the subsequent level (Appendix section A.2.2, Table 2).
> > > >
> > > > As desired by the reviewer, we will also add further analysis on the effect of increasing the number of HLT levels in CascadeXML architecture.

---

### Official Review · Reviewer_ZN1o · 2022-07-08

**Rating:** 7
**Confidence:** 4
**Soundness:** 3 good
**Presentation:** 3 good
**Contribution:** 3 good

**Summary:**

The authors propose CasccadeXML, an end-to-end deep learning pipeline for extreme multi-label text classification. CascadeXML leverages the multi-layer architecture of transformer models to handle different label resolutions with separate feature representations. Specifically, for each label resolution in the hierarchical label tree (HLT), features from the corresponding transformer layer are used to predict a shortlist of candidate labels or meta-labels. The authors empirically show that CascadeXML achieves state-of-the-art performance on three extreme multi-label datasets. The computation of training and inference are both faster than the current state-of-the-art transformer-based models.

**Questions:**

1. How did you decide the hyperparameters, especially the corresponding transformer layer of each label resolution?
2. Are the results sensitive to the hyperparameters?
3. Is there a rule of thumb for deciding these hyperparameters if given a new dataset?

**Limitations:**

The authors said the limitations and the potential negative societal impact are discussed in the appendix, but I didn't find them. Some discussion about the lack of theoretical guarantee or the limited search space of the hyper-parameters would be good.

**Strengths And Weaknesses:**

Strength
1. The approach to leveraging the multi-layer architecture of transformers for multi-resolution learning is interesting.
2. The experiment results are good concerning both performance and computation.
3. The analysis of the impact of label cluster size explains the improvement of CascadeXML well.
4. The paper is well-organized and easy to follow.

Weakness
Major
1. The method requires much more hyper-parameters, such as the corresponding transformer layer and the number of meta-labels of each label resolution.
2. Lack the discussion of the sensitivity of the newly introduced hyper-parameters.
Minor
3. The high/low resolution mentioned in Section 1 sounds ambiguous to me. May replace them with refined/coarse.
4. The definition of \mathfrak{C}^{(t)} and \mathfrak{R}^{(t)} are provided but the definition of \mathfrak{C}^{(t)}(y) and \mathfrak{R}^{(t)}(y) are not.
5. Line 234: Attention -> AttentionXML

---

> ### Author Response · Authors · 2022-08-02
> **response to reviewer ZN1o**
>
> 1. **Hyperparamters setting** :
>
> 1a. *Transformer layers and HLT level* : In CascadeXML, the last (12th) transformer layer always corresponds to the actual labels for the extreme task. In principle, layers 11th/10th/9th can be mapped to the three levels in the HLT with decreasing resolution. In practice, we found that skipping a transformer layer, so that embeddings at these layers are substantially different and hence more informative for the corresponding meta-classification tasks, gives some improvement. For visualization, please refer to Figure 1 in the main paper, and appendix section A.2.2 in the appendix for concrete mappings.
>
> 1b. *Number of meta-labels* : Choosing the number of meta-labels is a design decision which every HLT based XMC algorithm needs to make, see for instance Table 7 in XR-Transformer (https://arxiv.org/pdf/2110.00685.pdf, NeurIPS 2021).
>
> 2. **Sensitivity to hyperparameters** : The impact of varying the number of clusters in the penultimate layer is given in Table 4 of the paper. In the revised version, we will also add an ablation on varying the mapping between transformer layers and HLT levels.
>
> 3. **Terminology and Notation** : This terminology for “high/low resolution” is used to be in sync with the title of the paper. The definitions of $\mathfrak{C}^{(t)}(y)$ and $\mathfrak{R}^{(t)}(y)$ have been added, and "AttentionXML" has been fixed in the new uploaded version. Thanks for pointing out.

---

> > ### Comment · Reviewer_ZN1o · 2022-08-06
> > **Thank the authors for the response**
> >
> > My questions are well addressed and the revision looks good to me. Thank the authors for the response.

---

### Official Review · Reviewer_EmPK · 2022-07-11

**Rating:** 7
**Confidence:** 2
**Soundness:** 3 good
**Presentation:** 4 excellent
**Contribution:** 3 good

**Summary:**

The paper proposes CascadeXML to solve the extreme multi-label text classification problem. Compared to prior works, the proposed method utilizes features from multiple layers from Bert for different granularity of the hierarchical label tree. The coarser labels with lower scores can be terminated early for more efficient learning. The proposed method is compared with several SOTA methods, including  LightXML and XR-Transformer, on benchmark datasets - Wiki-500K, Amazon-670K, and Amazon-3M.

**Questions:**

Please refer to the weakness section

**Strengths And Weaknesses:**

I am not an expert in text classification, and thus not familiar with most of the related works. My judgment here is mostly based on the merits and the presentation of the paper on its own.

The paper in general is very well written. It explains the main drawbacks of existing works (although I am not sure if there are other works that already addressed these issues) and motivates from the drawbacks to propose CascadeXML. The core idea of CascadeXML is intuitively sound, well explained, and should be easy to reproducible. The authors also include the code in the supplementary.

I did not find too many weaknesses in the paper, but I do have some questions.

1. Does all compared (or the main) SOTA methods based on the same Bert model? This is not clearly stated in the paper.
2. For early termination of the coarse labels, there are risks of early wrong predictions as well.  Is there any measurement to prevent that?

---

> ### Author Response · Authors · 2022-08-02
> **response to reviewer EmPK**
>
> 1. **Sota methods** : All transformer-based approaches in Table 2 (LightXML, X-Transformer and XR-Transformer) use the same model ensemble set up as mentioned in the caption of Table 2. The model set up of baselines have been discussed in detail in the Appendix section A.2.1.
>
> 2. **Termination for coarse labels** : As a result of the candidate meta-label shortlisting procedure (also known as beam-search), early predictions of coarse/meta labels typically have high accuracy (and hence high recall of the actual labels, see Table 7 in Appendix B). We keep a shortlist size which not only ensures a high recall but also enough hard-negatives for the model to train efficiently.

---

### Official Review · Reviewer_2RC4 · 2022-07-13

**Rating:** 7
**Confidence:** 5
**Soundness:** 3 good
**Presentation:** 3 good
**Contribution:** 3 good

**Summary:**

This paper studies the extreme multi-label text classification problem and established new state-of-the-art (SOTA) results with the proposed method (namely CascadeXML), which significantly simplified the multi-stages training of the previous SOTA model, XR-Transformer. CascadeXML enables end-to-end training with two key insights: (1) leverages intermediate representations of Transformer layers for learning XMC sub-problems of the label tree (2) fine-tuning at the full resolution of label space.

**Questions:**

### Major Questions
- How to decide the mapping between Transformer intermediate layer embedding \theta^{(t)} and the level of the label tree? For example, if the depth of a label tree is greater than 12 layers (say 15 layers), where the BERT-base model has 12 layers of Transformer encoders, then the mapping is not 1-to-1?
- For Eq(4), It seems to be that the loss function is binary cross entropy. Any comments or justification on that? Did you compare it with other loss functions such as max margin hinge loss or ranking loss?
- Any thoughts of why the inference time is faster than other baselines in Table 3?
- For the CascadeXML trained by sparse-dense concat features, do you retrain the label tree or is it the original label tree constructed by PIFA of sparse TFIDF input features?
- It seems like the "coarsening-op" R_{\uparrow}^{(t)}(S) (latter one in Eq(1)) is never used in the main paper. Maybe you can consider removing it for more simpler presentation and ease of notation?

### Minor Comments
- Line 111: we often write t_k \in \mathcal{V} in convention.
- Line 181-182: For Algorithm 1, the short list at t+1 layer should be R_{\downarrow}^{(t)} (S_{top}^{(t)}) \union R^{(t+1)}(y). In Algorithm 1, the “refining-op” is missing?
- Line 327&332: merge duplicate bibtex entry [6] and [8], which points to the same paper.
- Line 432-433: update citation bibtex with latest info (Xiong et al., NAACL 2022)
- Line 446-447: update citation bibtex with latest info (Yu et al., JMLR 2022)



**Limitations:**

No potential negative societal impact.

**Strengths And Weaknesses:**

### Strength
- Leverage intermediate representations for multi-resolution learning of XMC is novel
- Paper writing is clear and easy to follow
- Empirical results are quite promising

### Weakness
- Lack of ablation study for the choice of intermediate layer of Transformer embeddings
- Lack of ablation study on loss functions

---

> ### Author Response · Authors · 2022-08-02
> **response to Reviewer 2RC4**
>
> **Answer to major questions**
> 1. **Mapping between Transformer layers and HLT level**: Please also check the common response above **Transformer layers and HLT levels** for our response on this question.
>
> 2. **Loss function** : Yes, for our experiments we have used BCE loss as this is the most commonly employed loss function in deep learning based XMC algorithms, and readily available in Pytorch library. As the focus of the paper is towards finding a more optimal architectural design, using the commonly employed loss functions enables a fair comparison. However, as suggested by the reviewer, we will add a comparison with the suggested loss functions in the later version of the paper.
>
> 3. **Inference time** : We attribute fast inference time to our multi-resolution dynamic hard-negative mining technique. Ideally, the dynamic approach is supposed to be fast, however, the implementation as introduced in LightXML is a much sub-optimal one. For our paper, we significantly improve on the implementation of dynamic hard-negative mining. With our code release, we will also release a faster implementation of LightXML for community usage.
>
> 4. **Label tree training** : CascadeXML is a purely transformer based approach and does not involve incorporating sparse TF-IDF input features in its base algorithm (Algorithm 1). For a fair comparison, however, we instead leverage DiSMEC as a post processing step to incorporate TF-IDF features along with our trained dense embeddings. DiSMEC does not require any negative sampling or use of a label tree. Please refer to Appendix Section A.4 for a detailed explanation.
>
> 5. **Coarsening** : Thank you for pointing this out, the “coarsening-op” is used to define R^{(t)}(y) which we had earlier missed out by mistake. We have updated it in the revised version of the paper in line 171.
>
> We have also fixed the points raised in minor comments in the updated version of the paper.

---

### Author Response · Authors · 2022-08-02
**common response**

We thank the reviewers for their comments. Please find below our responses starting with a common point raised by reviewers 2RC4 and zXmp. If required, we would be happy to clarify further, and look forward to a lively and insightful discussion.

**Transformer layers and HLT levels (Reviewer 2RC4 and zXmp)** : Since the original label tree is constructed as a binary tree, it has floor(log_2 L) levels, which is typically much greater than 12. For instance, for L = 3,000,000 in Amazon-3M, floor(log_2 L) is 21. However, as noted in the lines 153-155 of the paper (last line of Section 2), out of these, CascadeXML selects T (= 3 or 4) levels, each of which is mapped to one of the Transformer intermediate layers. This is also shown in Figure 1 of the paper, and Table 2 in Appendix gives the precise details of the mapping between HLT levels and the corresponding layer in the Bert model.

It may be noted that selecting fewer than floor(log L) levels for intermediate meta-classification is common in many recent tree-based XMC algorithms (lines 33-37) such as AttentionXML (3-4 intermediate levels), LightXML (1 intermediate level), and XR-Transformer (3-4 intermediate levels). The main difference between CascadeXML and other approaches is the following : while other approaches use the same final-feature embeddings of transformers as a blackbox for (meta-)classification tasks across selected HLT levels, CascadeXML, on the other hand, uses separate intermediate embeddings of the transformer architecture for each selected HLT level.

Also, note that we also provide visual analysis in Appendix B which corroborates our interpretation that different attention- and feature maps are needed for classification at different granularities of the label tree.

---

### Meta-Review · Area_Chair_stQk · 2022-08-23

**Recommendation:** Accept
**Confidence:** Certain

**Metareview:**

This paper proposes CascadeXML, which is an end-to-end framework for the task of tree-based extreme multi-label text classification. It extracts the representations from different layers of a BERT model and then maps them to different levels of the hierarchical label tree (HLT).

The proposed method shows strong performance in  P@k on benchmark datasets with an  improved efficiency during inference compared to other state-of-the-art methods including XR-Transformer and LightXML. Two of the reviewers pointed out the lack of ablation studies for the choice of intermediate layers of the BERT model and the mapping between those Transformer layser to the HLT layers. The problems were addressed in the updated version of the paper during rebuttal and the reviewers increased their scores as a result.

Given that 3 out of the 4 reviewers give a score of 7,  the recommendation is to accept the paper.


**Award:**

No

---

### Decision · Program_Chairs · 2022-09-14

Accept